# A lipoprotein partner for the *Escherichia coli* outer membrane protein TolC

Jim Horne[1†], Elise Kaplan[1†‡], Ben Jin[1], Kieran Abbott[2], Victor Flores[1], Emmanouela Petsolari[1], Jan Gradon[1§], Yvette Ntsogo[1], Andrzej Harris[1], Dingquan Yu[1], Ashraf Zarkan[2], Ben F Luisi[1*]

[1]Department of Biochemistry, University of Cambridge, Cambridge, United Kingdom; [2]Department of Genetics, University of Cambridge, Cambridge, United Kingdom

*For correspondence:
bfl20@cam.ac.uk

[†]These authors contributed equally to this work

Present address: [‡]Molecular Microbiology and Structural Biochemistry (MMSB), University of Lyon, Lyon, France; [§]School of Biochemistry, Faculty of Life Sciences, University of Bristol, Bristol, United Kingdom

Competing interest: The authors declare that no competing interests exist.

## eLife Assessment

In this **fundamental** work Horne et al present **compelling** evidence that YbjP is a novel binding partner of the TolC channel protein. The YbjP is characterized using cryo-EM, and its role probed using pull-down experiments, in vivo crosslinking, functional assays along with phylogenetic analysis which are all properly performed and presented and support the main conclusions. While the study does not identify a clear role for this protein, the revised manuscript offers improved clarity and contributes invaluable insight into membrane transport and antimicrobial resistance.

**Abstract** The outer membrane protein TolC from *Escherichia coli* belongs to an extensive super-family whose members are found throughout the didermal, Gram-negative bacterial lineages. The protein serves as an activated exit duct in multi-drug efflux pumps and protein secretion machinery. Many TolC homologues bear a lipid modification on the N-terminus that embeds into the inner leaflet of the outer membrane and appears to have been a conserved feature; however, the moiety is absent entirely in the *E. coli* TolC. We have discovered that the *E. coli* lipoprotein YbjP interacts extensively with the periplasmic surface of TolC and its N-terminal lipid moiety is embedded in the membrane, mimicking the intramolecular and modification-membrane interactions seen in TolC homologues. Here, we present cryo-EM structures of the MacA-MacB-TolC and AcrA-AcrB-TolC tripartite pumps complexed to YbjP. Although the association occurs spontaneously both in vitro and in vivo, the YbjP-TolC interaction is not required for efflux activity under standard laboratory conditions. YbjP may contribute to stabilising the orientation and distribution of TolC in the outer membrane, as well as the expression of transporters for tryptophan and cyclic peptide toxins.

## Introduction

The bacterial cell envelope is a highly complex compartment and protective barrier against the external environment, ensuring survival in response to chemical threats while maintaining individual cellular identity and communication of signals within communities. In Gram-negative species, including clinical pathogens such as *Escherichia coli*, *Pseudomonas aeruginosa*, *Haemophilus influenzae*, and *Helicobacter pylori*, the envelopes are characterised by an organisation of two lipid bilayer membranes and an interstitial partition, referred to as the periplasm, that encompasses a peptidoglycan sacculus conferring mechanical robustness (*Guest and Silhavy, 2023*; *Gumbart et al., 2021*). The envelope is home to a diversity of dedicated machineries that selectively move molecules across the barrier, facilitated by energy transduction. Harmful compounds such as antibiotics and bactericidal agents, as well as effector proteins, are displaced through this barrier by energy-dependent machines, and representative and well-characterised examples are the tripartite assemblies AcrA-AcrB-TolC and

MacA-MacB-TolC of *E. coli*, which are powered respectively by proton motive force and ATP binding and hydrolysis (*Du et al., 2018*). The AcrA-AcrB-TolC assembly drives efflux of chemically diverse compounds, and MacA-MacB-TolC can transport macrolides, peptide toxins, and host antibacterial peptides (*Honeycutt et al., 2020*; *Zhang et al., 2025*). Structures of these and analogous assemblies have provided insight into how the machines operate to recognise and move complex transport substrates in a preferred direction against concentration gradients (*Du et al., 2018*; *Fitzpatrick et al., 2017*; *Glavier et al., 2020*; *Tsutsumi et al., 2019*).

The outer membrane component of the efflux assemblies, TolC, is a homotrimeric protein with a self-closing β-barrel topology that inserts into the outer membrane and a long helical structural domain that interacts with periplasmic partners such as AcrA or MacA. TolC homologues are found throughout diderm bacteria, highlighting the functional importance of this superfamily (*Stubenrauch et al., 2022*). The long helical portion of TolC can be subdivided into an α-helical barrel domain, which appears to be a stable structural unit (*Calladine et al., 2001*; *Huang et al., 2014*), and conventional coiled-coil segments that dilate from a sealed resting state to an open state when interacting with a periplasmic partner (*Wang et al., 2017*). At the junction of the two helical regions, the polypeptide meanders to form a circumferential ring known as the equatorial domain that is proposed to interact with the peptidoglycan layer of the periplasm (*Shi et al., 2019*).

Here, in our analysis of the MacA-MacB-TolC tripartite assembly, we observed unexpected cryo-EM density spanning between the equatorial domain of TolC and the transmembrane boundary. Using a structural topology pattern recognition query against predicted folds of *E. coli* proteins, we identified the protein as a lipoprotein of unknown function, YbjP (UniProt ID P75818). The N-terminus of YbjP trails over the surface of TolC to reach the outer membrane where its lipid moiety is embedded. Our structural comparisons, together with functional and proteomic results from YbjP deletion strains, do not support an allosteric or signalling role for the lipoprotein in TolC-mediated efflux systems but suggest that YbjP is required for expression of transporters, including TnaB tryptophan permease and YojI ABC transporter that exports toxins such as microcin J25 through TolC (*Baquero et al., 2024*; *Delgado et al., 2005*).

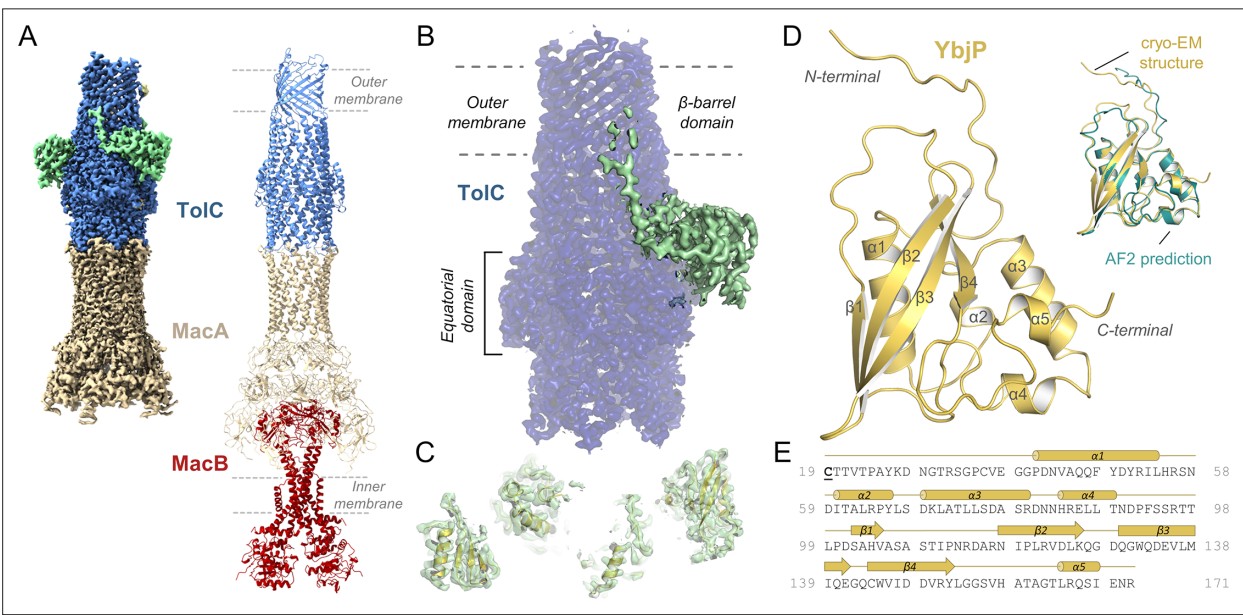

**Figure 1.** TolC partner identified by cryo-EM. (**A**) Cryo-EM density map (left) resolved here and model (right) for the tripartite pump MacA-MacB-TolC (PDB 5NIL). Density for the MacB dimer is less well defined. (**B**) Close-up view on the TolC region showing additional density in green. TolC domain organisation is indicated. (**C**) Modelling of secondary structural elements fitting in the additional density. (**D**) The refined structure of the TolC partner, YbjP. After lipoprotein maturation, the N-terminal cysteine carries a lipid modification which is not shown here. Alignment of the YbjP structure from cryo-EM and corresponding AlphaFold prediction model is shown as inset. (**E**) Sequence and secondary structure alignment annotation of mature YbjP. The tri-acylated cysteine (C19) conserved in lipoproteins is underlined.

The online version of this article includes the following figure supplement(s) for figure 1:

**Figure supplement 1.** Cryo-EM processing workflow for MacAB-TolC-YbjP complex.

## Results

### Cryo-EM structure of TolC bound to a lipoprotein

Earlier cryo-EM work on the MacA-MacB-TolC assembly used samples reconstituted in detergents (*Fitzpatrick et al., 2017*). We explored the preparation of MacA-MacB-TolC in peptidiscs as a better mimic of the membrane environment (*Carlson et al., 2018*). From these novel preparations, we were able to solve the structure by cryo-EM with interpretable density for all three components of the assembly (*Figure 1A*). In one of our preparations, additional density was observed near the equatorial domain of TolC, extending towards the outer membrane β-barrel domain (*Figure 1B*). We refined this structure to a final resolution of 2.48 Å (*Figure 1—figure supplement 1* and *Supplementary file 1*) and fitted the additional density with secondary structural elements (*Figure 1C*). The incomplete candidate fold was then searched against the database of *E. coli* protein structures predicted by AlphaFold2 (*Jumper et al., 2021*) using the DALI topology recognition programme (*Holm, 2022*). The result matches the lipoprotein YbjP (P75818) which could be fitted into the cryo-EM map. The refined model is presented in *Figure 1D*. The lipoprotein globular domain is formed by a series of four consecutive α-helices (α1-α4), followed by four antiparallel β-strands (β1-β4) and a C-terminal α-helix (α5) (*Figure 1E*). Structural alignment of the predicted YbjP and cryo-EM refined models reveals a strikingly high degree of similarity (RMSD of 0.53 Å over 118 residues), confirming the accuracy of the prediction (*Figure 1D*, inset). A superposition of the MacAB-TolC-YbjP complex with our previously reported MacAB-TolC structure (*Fitzpatrick et al., 2017*) revealed that TolC and the tripartite pump are largely unaffected upon YbjP binding with an RMSD of 1.03 Å over 1258 residues for TolC trimer (*Figure 2A*). Nonetheless, the side chains of a few residues in TolC, which mainly correspond to positively charged amino acids (R18, R24, K214, R227, R234), reorient to interact with the YbjP lipoprotein partner (*Figure 2B*).

### Interactions between TolC and lipoprotein YbjP

YbjP straddles two adjacent TolC protomers, and the interface is stabilised by multiple hydrogen bonding and ionic interactions involving the linker and globular domain of YbjP and two TolC subunits (*Figure 2C*). The intermolecular contacts engage almost 20 residues in both TolC and YbjP. The lipoprotein flexible linker runs along TolC N-terminal helix from the equatorial to the transmembrane domains and is stabilised by several hydrogen bonds involving N32, R35, R24, Y98, and L17 from one TolC monomer (*Figure 2C*, panel 1). This cluster is consolidated by hydrogen bonding interactions between S107, S109, T110, and N113 from YbjP loop connecting beta-strands β1 and β2 and R234, Q306 and Y307 from the proximal adjacent TolC monomer, maintaining this flexible loop perpendicular to the lipoprotein linker. YbjP D115 and N118 also make contacts with R18 and E314 from TolC.

Interestingly, the YbjP globular domain is strongly associated via a cluster of ionic interactions (*Figure 2C*, panel 2), mediated by a trio of positive, negative, positive residues (R122, E135, R151) which each contacts a residue of opposite charge in TolC (D231, R227, E317). Finally, numerous hydrogen bonds are formed between the YbjP globular domain and helices from both TolC monomers (*Figure 2C*, panel 3) engaging G155, V157, and A159 backbones, and K70, N118, E135, and R151 side chains from the lipoprotein. Residue conservation of YbjP was evaluated visually with the server CONSURF, revealing that the conserved residues are located at the interface with TolC (*Figure 2D*).

Like other lipoproteins, YbjP is predicted to be processed by cleavage of the signal sequence at residue C19 (also referred to as the +1 Cys of the matured lipoprotein), then lipidated as N-palmitoyl and S-diacylglycerol cysteine. These processing and modification steps are likely to occur at the inner membrane (*Guest and Silhavy, 2023*). Although the density is not well defined for the lipid moiety due to its flexibility, the map does reveal a portion of the hydrocarbon embedded into the inner leaflet of the outer membrane. This is consistent with mass spectrometry studies that have identified YbjP as an outer membrane protein (*Molloy et al., 2000*). Interestingly, there is no density for the lipoprotein R3 acyl chain, suggesting that the cysteine α-amino group has not been modified, while *E. coli* lipoproteins are presumed to be tri-acylated to traffic via the Lol system (*Grabowicz and May, 2026*; *Noland et al., 2017*). Further characterisation is needed to establish if YbjP is naturally di- or tri-acylated.

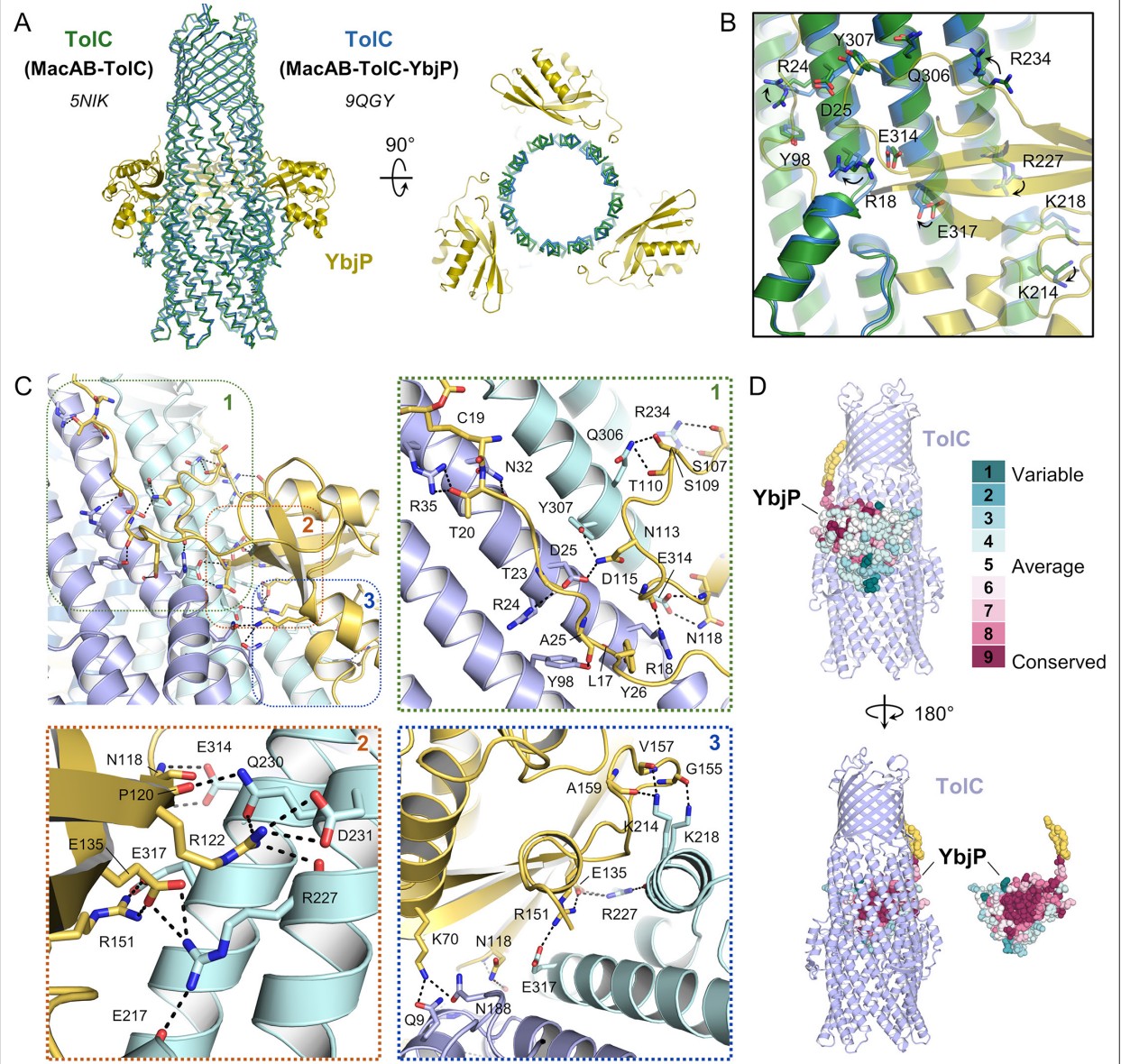

**Figure 2.** Interactions in the TolC-YbjP complex. (**A**) Superposition of TolC from the MacAB-TolC-YbjP assembly reported this paper in blue (TolC) and yellow (YbjP) (PDB 9QGY) and from the previously reported MacAB-TolC complex in green (PDB 5NIK). (**B**) Close-up view on the YbjP-TolC interface with TolC residues involved in the interaction shown as sticks. (**C**) Overview of protein-protein interactions with close-up views presented in panels 1, 2, and 3. The YbjP lipoprotein (yellow) contacts two adjacent protomers of the TolC trimer (blue). Lipoprotein acyl modification and residues involved in intermolecular contacts are shown in stick representation. (**D**) Residue variability analysis of YbjP highlighting the clustering of conserved residues at the TolC interface. Analysis was performed with the CONSURF server. YbjP is shown as spheres with the lipidation in yellow.

## Validation of the TolC-YbjP interaction by in vitro pull-down and in vivo photo-crosslinking

To validate the TolC-YbjP interactions, we used an in vitro affinity pull-down assay with purified proteins. For this, we produced a soluble mutant of YbjP, hereafter referred to as YbjP$_s$, removing the lipobox conserved cysteine that carries the acyl chains, and the eight following residues corresponding to a flexible linker joining the acylated cysteine to the protein globular domain. The protein (D28-R171) with an N-terminal His-tag was purified from the cytoplasm and immobilised on nickel resin. As shown in *Figure 3A*, detergent-purified TolC remained attached to the resin only in the presence of YbjP$_s$, confirming the association between the two proteins and demonstrating that the lipoprotein lipid moiety is not required for the interaction to occur. We could not detect an interaction

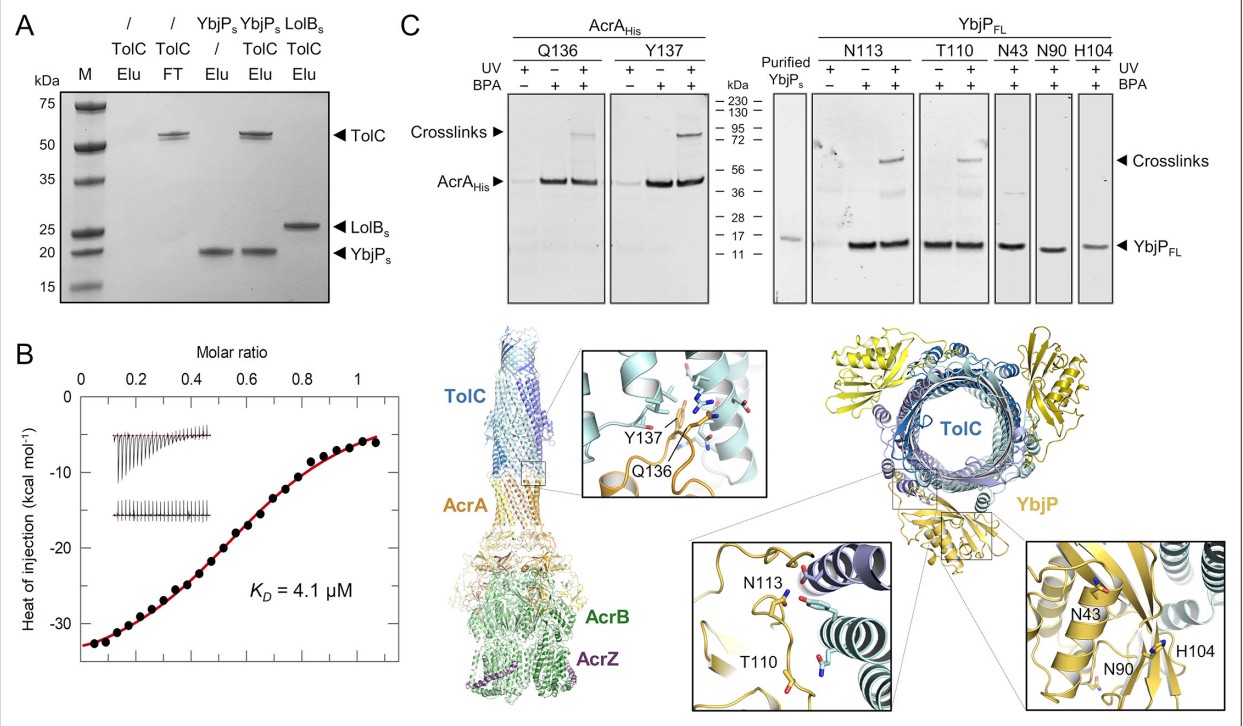

**Figure 3.** In vitro and in vivo validation of the TolC-YbjP interface. (**A**) In vitro binding assay between TolC and soluble YbjP that is missing the lipid modification. TolC-FLAG was mixed with N-terminally His-tagged soluble YbjP (YbjP$_s$) or soluble LolB (LolB$_s$) prior to immobilisation on immobilised metal affinity chromatography (IMAC) resin. After several washes, the elution fractions (Elu) were analysed by SDS-PAGE. Molecular masses of protein standards (**M**) are indicated. (**B**) Isothermal titration calorimetry (ITC) profile for the interaction of TolC and YbjP$_s$. Background-corrected heats of injection are shown together with a fitted binding curve in red. The raw thermograms corresponding to the injection of YbjP$_s$ into a TolC-containing cell or into buffer are shown in upper and lower insets, respectively. ITC parameters are listed in **Supplementary file 2**. (**C**) In vivo validation of the interaction between YbjP$_{FL}$ and TolC by photo-crosslinking. The experimental procedure was validated using another periplasmic lipoprotein, AcrA, which associates with TolC. As shown underneath, AcrA residues Q136 and Y137, proximal to TolC in the structure of the AcrABZ-TolC pump (PDB 5NG5), were replaced by pBPA. For YbjP, the two residues N113 and T110 proximal to TolC in the MacAB-TolC-YbjP complex (PDB 9QGY) and the three residues N43, N90, and H104 distal to TolC were mutated. In brief, E. coli C43 cells expressing YbjP$_{FL,His}$ or AcrA$_{His}$ mutants carrying a photoactivatable pBPA group at indicated positions were grown in the presence (+) or absence (–) of pBPA and irradiated (+) or not (–) with UV. Cells were then lysed by successive freeze-thaw cycles, solubilised with DDM and lysates purified by Ni-NTA chromatography. Eluted proteins were analysed by SDS-PAGE and immunoblotting using anti-His antibodies. Purified YbjP$_s$ was loaded as a benchmark control. Presence of crosslinked products is indicated. Note that FL YbjP migrates faster than its predicted molecular weight (~19 kDa), likely due to the presence of the lipoprotein lipid chains. A double detection of YbjP$_{FL,His}$ and endogenous TolC proteins for the N113 sample is shown in **Figure 3—figure supplement 1**.

The online version of this article includes the following source data and figure supplement(s) for figure 3:

**Source data 1.** Uncropped gels for **Figure 3** without labels.

**Source data 2.** Uncroppsed gels for **Figure 3** with labels.

**Figure supplement 1.** Double detection of YbjP and TolC proteins for the in vivo photo crosslinking.

**Figure supplement 1—source data 1.** Uncroppsed gels for **Figure 3—figure supplement 1** without labels.

**Figure supplement 1—source data 2.** Uncropped gels for **Figure 3—figure supplement 1** with labels.

between YbjP$_s$ and Pal (peptidoglycan-associated protein) lipoprotein or between Pal and TolC using the pull-down assay.

Interactions between TolC and YbjP$_s$ were also measured by isothermal titration calorimetry (**Figure 3B** and **Supplementary file 2**). The binding event is exothermic, consistent with electrostatic interactions at the protein-protein interface. The dissociation constant is in the micromolar range ($K_D$ ~4 µM). Although this value represents a weak to moderate association, the affinity is likely to be substantially higher in the presence of full-length YbjP which includes the 8-residue linker connecting the lipid modification to the globular domain. This region is absent here but participates in favourable interactions with TolC and is likely to increase the affinity (**Figure 2C**, panel 1). Additionally, the lipid

modification would localise YbjP to the membrane, reducing its configurational entropy and thereby increasing TolC binding.

To validate that YbjP and TolC interact within the cell, we engineered C-terminally His-tagged full-length YbjP (YbjP_FL) to contain the photo-crosslinkable amino acid para-benzophenone (pBPA) (*Chin et al., 2002*; *Young et al., 2010*) at positions guided by the YbjP-TolC structure. We replaced residues at the protein-protein interface (T110, N113) or at a distance (N43, N90, and H104) as negative controls. To validate the experiment, we also constructed amber mutants of His-tagged AcrA, an inner membrane protein known to interact with TolC, and substituted Q136 and Y137, two residues proximal to TolC in the AcrABZ-TolC efflux pump structure (*Wang et al., 2017*). Cells were grown in a medium supplemented with pBPA, induced and then irradiated for 15 min with ultraviolet light. After cell lysis, His-tagged proteins were retrieved from the samples by affinity chromatography. Purified fractions were analysed by SDS-PAGE and immunoblotting (*Figure 3C*). We observed covalent photo-adducts that migrate at higher molecular weight compatible with AcrA-TolC or YbjP-TolC adducts only for the pBPA-supplemented cells and only after illumination with UV light for the residues that are proximal to the TolC surface, but not those that are distal. We confirmed the presence of both YbjP and TolC in the N113 YbjP crosslinked product by western blot (*Figure 3—figure supplement 1*). These results are consistent with a model for YbjP lipoprotein interactions with TolC in vivo.

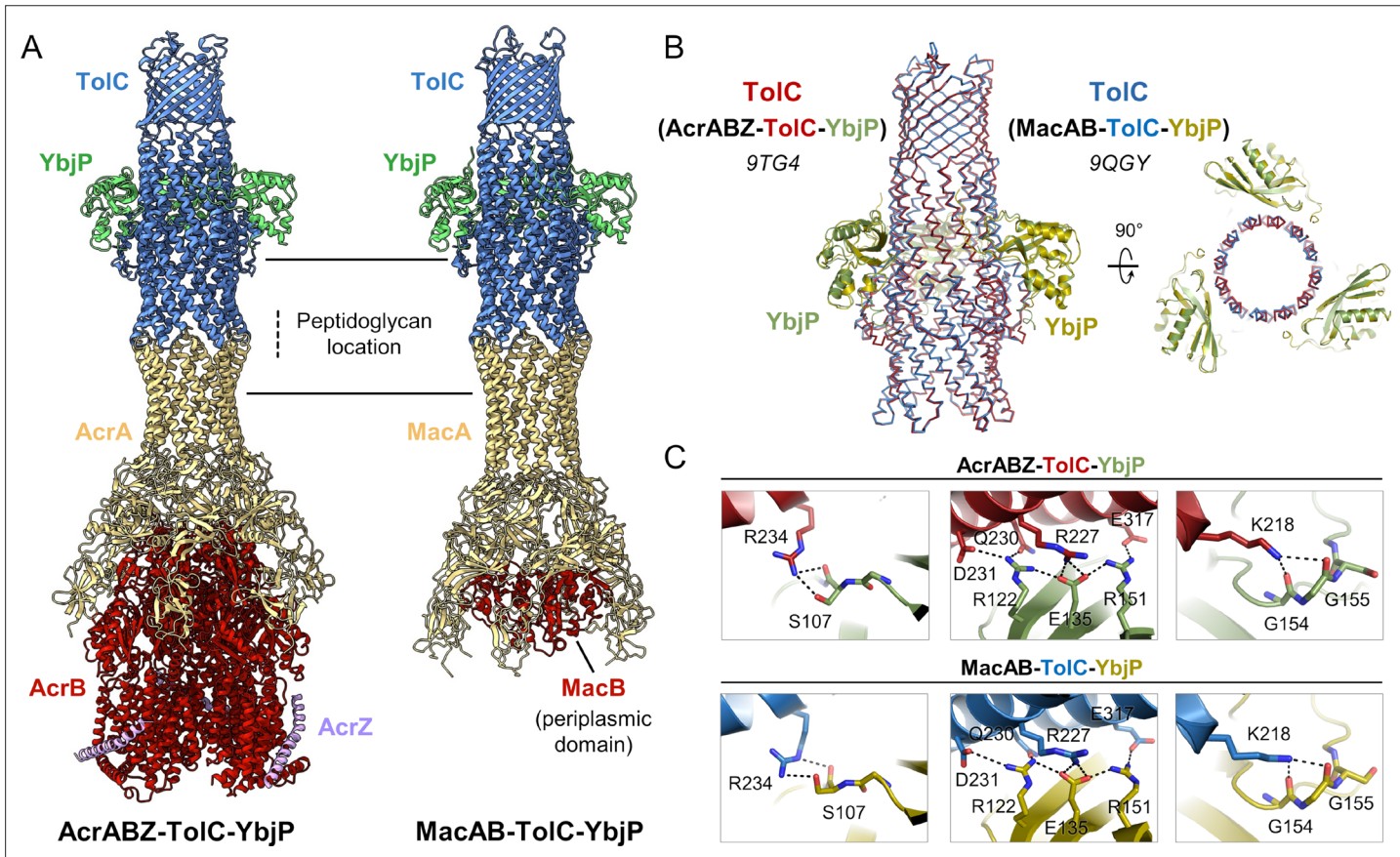

**Figure 4.** The structural comparison of AcrABZ-TolC-YbjP and MacAB-TolC-YbjP. (**A**) Cryo-EM maps and models of AcrABZ-TolC-YbjP and MacAB-TolC-YbjP reconstituted in peptidisc. (**B**) Superposition of TolC from the AcrABZ-TolC-YbjP and MacAB-TolC-YbjP assemblies reported here (red: TolC, green: YbjP, PDB 9TG4; blue: TolC; and yellow: YbjP, PDB 9QGY). (**C**) Close-up views on the YbjP-TolC interface in the AcrABZ-TolC-YbjP (top) and MacAB-TolC-YbjP (bottom) structures.

The online version of this article includes the following figure supplement(s) for figure 4:

**Figure supplement 1.** Cryo-EM workflow for AcrABZ-TolC-YbjP complex.

**Figure supplement 2.** YbjP binds at the same position in the AcrABZ-TolC and MacAB-TolC tripartite pumps and may not interact with peptidoglycan.

## TolC-YbjP interaction in the AcrA-AcrB-AcrZ-TolC assembly

To examine whether YbjP can interact with other TolC-mediated assemblies, we supplemented the AcrABZ-TolC pump reconstituted in peptidisc with YbjP$_s$ and resolved the cryo-EM structure of the complex at 3.17 Å resolution (*Figure 4—figure supplement 1* and *Supplementary file 1*). Structural comparison of the two assemblies reveals that YbjP binds at the same location on TolC (*Figure 4A and B*) and makes the same interactions (*Figure 4C*). The AcrABZ-TolC-YbjP assembly was also reconstituted into peptidoglycan (*Figure 4—figure supplement 2* and *Supplementary file 3*), and the position of the peptidoglycan layer between AcrA and TolC corresponds with the localisation previously inferred from low-resolution cryo-ET data (*Shi et al., 2019*; *Figure 4—figure supplement 2*). The reconstructed maps indicate that the YbjP may not interact with the peptidoglycan layer, but this cannot be certain due to the limited resolution of the cryo-EM map. In vitro characterisations indicated that YbjP can interact with peptidoglycan in pull-down assays, but this is likely to be non-specific as alanine substitutions in the domain proximal to the peptidoglycan (R80, D81, R85, and E86) had little impact on the efficiency of the pull-downs. Photo-crosslinking in substitutions of the photoreactive amino acid pBPA at positions D77, Q166, E169, and R171 did not give evidence for peptidoglycan-linked species. The data suggest that YbjP may not form avid interactions with the peptidoglycan.

## Evolutionary profile of YbjP

To explore the possible function of YbjP, we investigated its evolutionary profile. In the PFAM database, YbjP is annotated to contain one DUF3828 domain. The most well-characterised DUF3828-containing protein is Tai3, a periplasmic type VI cognate immunity protein of the T6SS amidase effector Tae3 (*Dong et al., 2013*; *Russell et al., 2014*). Another periplasmic protein YqhG contains one full and one partial DUF3828 domain and is involved in regulating type I fimbriae expression in *E. coli* by an unknown mechanism (*Bessaiah et al., 2019*). Structural alignment of YbjP with the Tai3 crystal structure and YqhG AlphaFold2 model indicates that these proteins share a common fold (*Figure 5A*).

To investigate whether YbjP is functionally homologous to Tai3 or YqhG, we reconstructed a phylogenetic history of the DUF3828-containing proteins, which could be classified into three major families based on taxonomy, gene synteny, presence of a signal peptide and a conserved QDX motif, involved in Tae3 inhibition (*Supplementary file 4*). The result indicates that the function of YbjP is likely distinct from that of Tai3 or YqhG (*Figure 5B*). We noted that Tai3 is found in a broader range of bacterial groups (Gammaproteobacteria, Betaproteobacteria, and Bacteroidota/Chlorobiota) compared to YbjP. Similarly, YbjP is restricted to some members of the Enterobacterales order within the class of Gammaproteobacteria, unlike TolC, which is more widespread (*Figure 5—figure supplement 1*). Notably, the surface of TolC corresponding to the TolC-YbjP interface is more conserved in Enterobacterales with YbjP than in those without (*Figure 5C*), suggesting a scenario where YbjP evolved recently from an ancestral Tai3-like protein to become a TolC interactor.

## YbjP is not required to compensate for lack of TolC lipidation

Outer membrane efflux proteins structurally homologous to TolC often carry an N-terminal acylation. This is the case for the drug exit duct OprM in *Pseudomonas* and the outer membrane efflux proteins in *E. coli* MdtP, MdtQ, and CusC, but with the notable exception of TolC. The reason for the presence or absence of acylation remains unclear, but it has been proposed that acylation has roles in anchorage to the membrane and in membrane insertion. We first postulated that YbjP could compensate for the absence of lipid modification in TolC. Indeed, YbjP linker and N-acetylation mimic the linker and acetylation arrangement seen in OprM and CusC (*Figure 6A*). Similarly, AlphaFold3 predicted the interaction of YbjP with TolC but not with the other TolC structural homologues that are lipidated in *E. coli* (*Figure 6B*). Consistent with this, we found that *E. coli* TolC is distinct from its lipidated structural homologues (*Figure 6C*), suggesting the possibility that YbjP evolved to compensate for the absence of lipidation. However, further bioinformatic analysis reveals that while most organisms containing the *ybjP* gene possess non-lipidated TolC, non-lipidated TolC is also present in most organisms lacking *ybjP* (*Figure 6D*). This indicates that YbjP may not be necessary to compensate for the lack of lipidation of TolC.

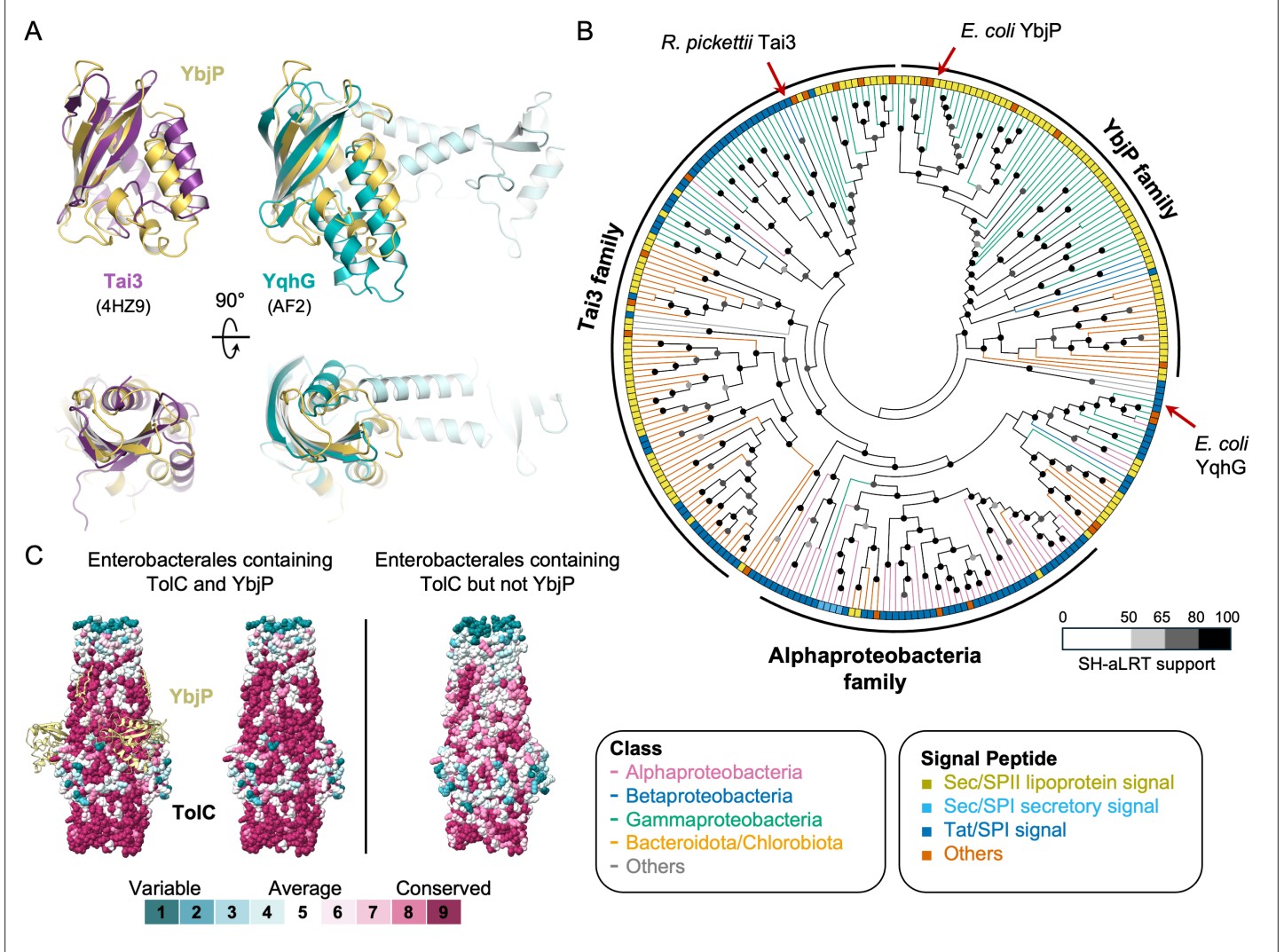

**Figure 5.** YbjP evolution and co-evolution with TolC. (**A**) Structural alignment of YbjP with other DUF3828-containing proteins: Tai3 (PDB 4HZ9, purple) and YqhG (AlphaFold2 Q46858, blue). A 90° rotation along the *x*-axis is shown underneath. (**B**) A cladogram of DUF3828-containing proteins. The reviewed proteins (Tai3, YbjP, and YqhG) are marked by red arrows. The signal peptide was predicted by SignalP6.0: yellow: Sec/SPII lipoprotein signal; light blue: Sec/SPI secretory signal; blue: Tat signal; orange: others. (**C**) Surface conservation of TolC proteins from Enterobacterales containing or not containing YbjP. Results are displayed on *E. coli* TolC on the left and on *Pectobacterium atrosepticum* TolC (AlphaFold model, Q6DAC5) on the right. For clarity, YbjP is displayed on the left panel but not on the right. Analysis was performed with the CONSURF server.

The online version of this article includes the following figure supplement(s) for figure 5:

**Figure supplement 1.** The cladogram of TolC (IPR010130) in *Pseudomonadota*.

**Figure supplement 2.** Structural comparison of TolC$_3$-YbjP$_3$ and TolC$_3$-SlyB$_{11}$.

## Exploring in vivo function of YbjP

To explore the physiological function of YbjP, we compared the proteomic profiles of Δ*ybjP*, Δ*tolC*, Δ*tolC* Δ*ybjP*, and parental strains under exponential and stationary phase conditions (*Figure 7*). Only minor differences were detected in the proteome of the Δ*ybjP* strain compared with the parental wild-type in both growth stages, while numerous significant changes were observed for the Δ*tolC* and Δ*tolC* Δ*ybjP* strains. Levels of TolC were not impacted in the Δ*ybjP* strain, indicating that YbjP is not required for TolC biogenesis. Previous RNA-seq analysis highlighted an upregulation of *ybjP* in response to membrane-disrupting chemicals, including vanillin (*Pattrick et al., 2019*) and EDTA (*Janssens et al., 2024*). However, we do not observe an impact on growth under standard laboratory conditions, as the minimal inhibitory concentrations for growth inhibition were unchanged in the

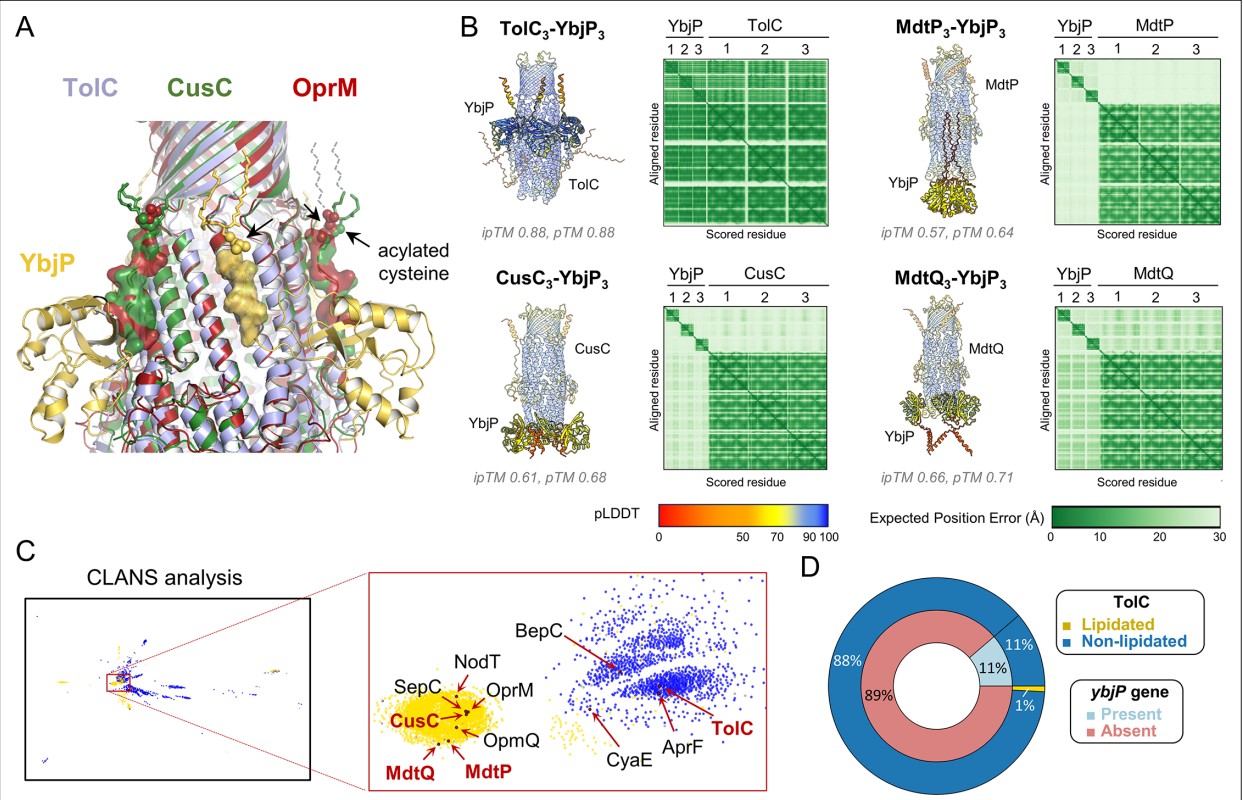

**Figure 6.** YbjP lipoprotein is not required for TolC trafficking in *E. coli*. (**A**) Structural alignment of *E. coli* TolC (blue), CusC (green, PDB 3PIK), and *P. aeruginosa* OprM (red, PDB 3D5K). The YbjP- or TolC-homologue linker is shown as a transparent surface, and modified cysteines are indicated by arrows. Dashed lines show the extension of the lipid group (partially- or non-modelled in the structures). (**B**) AlphaFold3 prediction (*Abramson et al., 2024*) between *E. coli* outer membrane efflux proteins and YbjP. The model with the highest ipTM and pTM scores is shown out of three technical repeats. (**C**) CLANS (CLuster ANalysis of Sequences) analysis of outer membrane efflux proteins (IPR003423) in *Pseudomonadota*. Clustering was performed in 2D until equilibrium. Outer membrane proteins in *E. coli* are shown in red, and other key reviewed proteins in black. Yellow, lipidation; blue, secreted; grey, other. (**D**) Distribution of the *ybjP* gene and lipidated TolC in Gammaproteobacteria. Repartitions are as follows: *ybjP* present (light blue, n=67) or absent (magenta, n=525); lipidated (yellow, n=5) or non-lipidated (dark blue, n=587) TolC.

*ΔybjP* compared to the parental strains in the presence of vanillin and EDTA; similarly, no changes were observed with benzalkonium chloride and different classes of antibiotics, under either aerobic or microaerobic conditions (*Supplementary file 5*).

Considering that YbjP expression is regulated by the stress response sigma factor *rpoS* (*Lacour and Landini, 2004*), its physiological role is likely context-dependent and not readily apparent under standard laboratory conditions. Indeed, the few proteins whose levels were influenced in the *ΔybjP* strain are associated with specific contexts, including stress and changes in lifestyle (*Figure 7*). We observed the downregulation of Antigen 43 (*flu*), which is an autotransporter porin with an extensive extracellular domain that is involved in auto-aggregation and which promotes the shift from a planktonic state to a sessile state. While previous work noted that YbjP overexpression influences motility (*Tenorio et al., 2003*), we did not observe any effect of the *ybjP* deleted strain on LB motility agar plates (*Supplementary file 6*).

In addition, changes in specific transporters were noticed. The YojI ABC transporter, proposed to be involved in the efflux of the microcin J25 toxin in conjunction with TolC, was decreased in relative abundance in the *ΔybjP* strain in stationary phase (*Figure 7*). However, we did not observe any change in MIC values for the *ΔybjP* strain in the presence of microcin J25 during exponential phase, while the *ΔtolC* and *ΔtolC ΔybjP* strains did show increased sensitivity of >4000-fold (*Supplementary file 5*). Conversely, the TnaB low-affinity tryptophan permease level increased. We did observe a small increase in growth for the *ΔybjP* strain compared to the parental strain in minimal media in the presence of tryptophan. Interestingly, the *ΔtolC* and *ΔtolC ΔybjP* strains did show marked decreased growth in the absence of tryptophan (*Figure 7—figure supplement 1*). The outer membrane pore

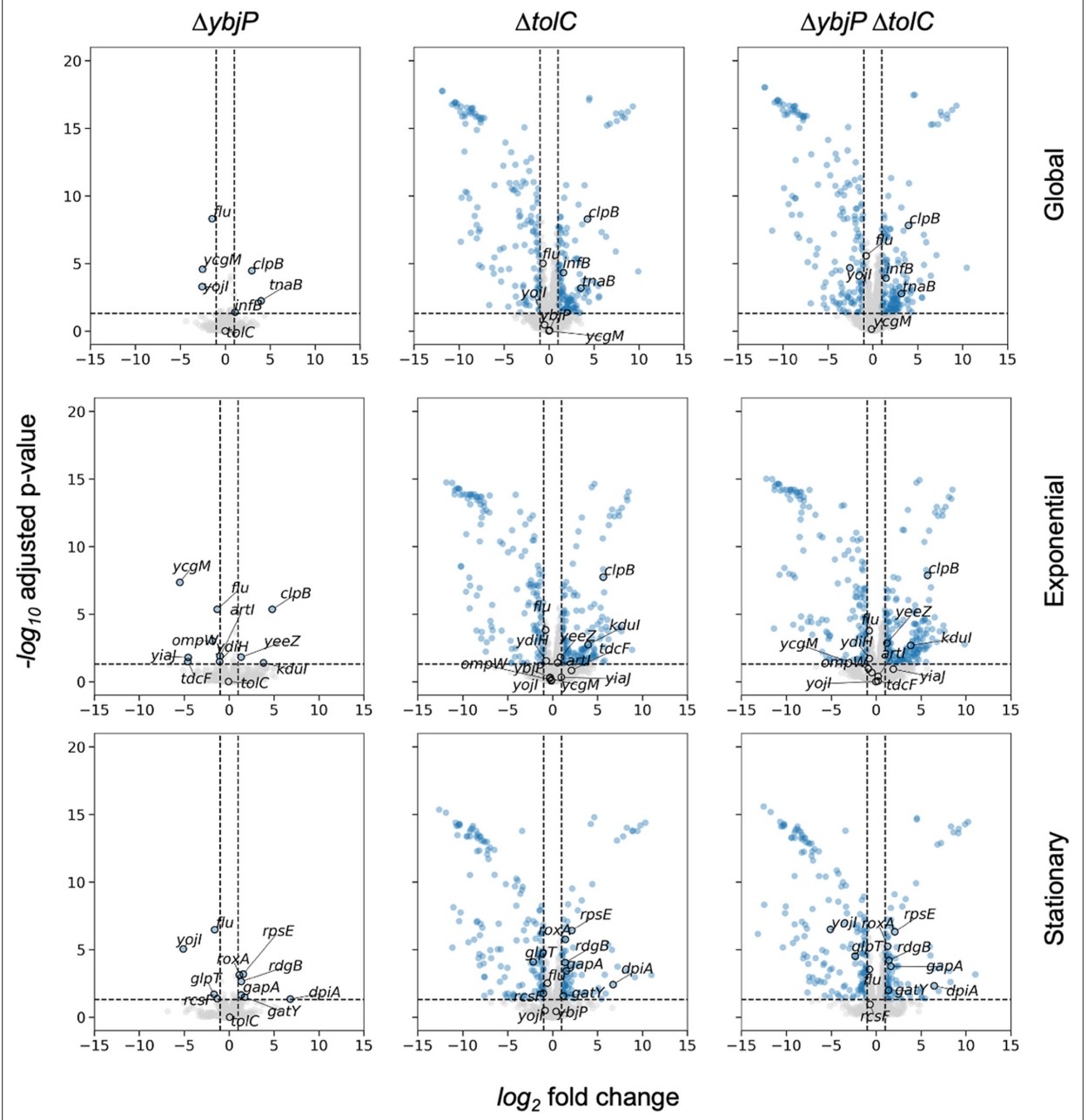

**Figure 7.** YbjP lipoprotein deletion has modest impacts on the proteome in *E. coli*. Volcano plots showing the proteome-wide differential expression relative to WT. Columns correspond to Δ*ybjP*, Δ*tolC*, and Δ*ybjP* Δ*tolC* (left to right), and rows correspond to the global, exponential, and stationary phases (top to bottom). Proteins with |log₂ fold change|>1 and Bonferroni-Hochberg adjusted p-value<0.05 are shown in blue. Significant hits in Δ*ybjP* and selected genes are labelled.

The online version of this article includes the following figure supplement(s) for figure 7:

**Figure supplement 1.** Impact of tryptophan on growth in minimal media.

OmpW, which plays a minor role in the efflux of quaternary cationic compounds together with the EmrE transporter (*Beketskaia et al., 2014*), was downregulated. Taken together, the changes in the proteome implicate nuanced, multiple roles of YbjP, but the phenotypic impact of the null strain is likely to be seen under environmental stress conditions.

## Discussion

Our structural and functional results indicate that the lipoprotein YbjP is located to the outer membrane, where it can interact with the membrane protein TolC. The mode of interaction between TolC and YbjP resembles closely the intramolecular interactions of the lipidated N-terminus in TolC paralogs, such as OprM from *Salmonella*. The interaction is likely functionally important within some Enterobacterales. It is proposed that the TolC archetype may bear sequence or environmental features that have limited its distribution, or it is a more recently evolved form in the TolC-like protein family (*Stubenrauch et al., 2022*). It is possible that TolC and YbjP co-evolved in this lineage.

The biogenesis of TolC presents intriguing mechanistic puzzles. The insertion stage of outer membrane proteins involves the β-barrel assembly machine (BAM) complex (*Voulhoux et al., 2003*; *Wu et al., 2005*), but the process is not well characterised for a non-canonical porin-like protein with an extended hydrophilic portion, such as TolC, nor are the earlier steps of translocation across the periplasm. The chaperone translocation and assembly machinery facilitates assembly of TolC into the outer membrane in a potentially redundant pathway (*Stubenrauch et al., 2022*). Reflecting on the potential requirements suggests key aspects to consider for understanding the process. First, the protomers must be moved across the periplasm from the translocon where the nascent chains emerge. Second, this insertion would be coordinated to deliver three protomers to fold into the full trimeric architecture. Lastly, the folding does not require exogenous energy and is likely driven by hydrophobic interactions with the highly crowded and asymmetric lipid bilayer (*Horne et al., 2020*).

Mechanistic puzzles also arise when considering how lipoproteins such as OprM and YbjP arrive at the outer membrane and are inserted. Lipoproteins are processed and lipid groups are attached to a conserved cysteine residue that immediately follows the signal sequence in the inner membrane (*Guest and Silhavy, 2023*). Acylated thiols are found in components of other bacterial efflux assemblies. The structure of CusC indicates the presence of an acylated thiol on the N-terminal cysteine, and acylation is also found in CmeC, but nothing has been found to be added to the N-terminal cysteine of the MtrE structure. Post-translational lipidation is particularly essential for the secretion and localisation of some membrane proteins, a process involving different acyl transferases (*Aicart-Ramos et al., 2011*; *Kovacs-Simon et al., 2011*; *Linder and Deschenes, 2007*; *Nakayama et al., 2012*). Why proteins that are embedded within cellular membranes via a large hydrophobic structural domain need modifications such as the N-terminal lipidation appears puzzling. *Akama et al., 2004*, suggested that these proteins must first be anchored to the membrane by an N-terminal lipid so that the insertion of their large hydrophobic domain may be triggered. Structural evidence indicates that this later step is critical for the correct folding of outer membrane factors (OMFs) and highlights the possible involvement of membrane-interacting components in the OMF opening process and consequently the efflux pump function (*Lei et al., 2014*).

Translocation of lipoprotein to the outer membrane in *E. coli* requires five essential proteins (LolA-E) (*Guest and Silhavy, 2023*). The mechanism involves lipoprotein extraction from the inner membrane by the ABC transporter LolCDE, then transfer to the periplasmic carrier LolA for delivery to the outer membrane acceptor LolB, which facilitates insertion into the outer membrane. It is envisaged that this pathway is involved in the movement of OprM and its lipidated outer membrane homologues. As *E. coli* TolC is not lipidated, it is possible that YbjP acts as a chaperone that intercepts the TolC and aids its translocation across the periplasm via the Lol system. However, this proposed pathway is likely redundant or not essential, as TolC is still found in the outer membrane in strains null for YbjP. The BAM complex impacts on TolC assembly in *E. coli* (*Malinverni et al., 2006*; *Werner and Misra, 2005*), but surprisingly it is not involved in assembly of the TolC-like proteins OprM in *P. aeruginosa* (*Hoang et al., 2011*). The pathway for insertion is likely to be redundant and divergent for the TolC family (*Stubenrauch et al., 2022*). Notably, a broad screen of cell envelope protein complexes in *E. coli* did not identify physical or genetic interactions between YbjP and TolC (*Babu et al., 2018*), but such interactions were found between YbjP and the partners of TolC in the tripartite assembly, namely the RND transporter AcrB and the periplasmic AcrA. It may be that the interactions between TolC and YbjP in vivo are weak and transient, as might occur, for example, during TolC insertion into the outer membrane.

Our results do not support an allosteric or signalling role for YbjP but suggest that the lipoprotein impacts on expression of transporters tryptophan permease TnaB and ABC family member YojI, which transports toxins such as microcin J25 through TolC (*Baquero et al., 2024*). It is also possible that

YbjP may facilitate the biogenesis of TolC during times of rapid adaptation or its distribution in the membrane. To our knowledge, YbjP is the first protein identified to interact with the lateral surface of TolC. Interestingly, the outer membrane protein SlyB has been shown to encapsulate TolC-like proteins during stress conditions, and cryo-EM 2D classification images predict the binding of SlyB at the TolC-YbjP interface (*Janssens et al., 2024*). Structural predictions from AlphaFold3 further support that SlyB binds TolC at the YbjP location (*Figure 5—figure supplement 2*). Given that both YbjP and TolC are upregulated in response to EDTA treatment (*Janssens et al., 2024*), YbjP and SlyB might compete for binding to TolC under conditions of outer membrane stress. Possibly, YbjP might support clustering and partitioning of transport machineries during cell division or stress (*Zhang et al., 2025*), which in turn can influence the efficiency of adaptation to environmental conditions.

## Materials and methods

### Preparation of disulphide-engineered MacAB-TolC and reconstitution into peptidisc

*E. coli* BL21 (DE3) strain was transformed with plasmids pET20b-hexaHis-MacA$_{D271C}$- MacB$_{G465C}$ expressing the MacA$_{D271C}$ and MacB$_{G465C}$ mutants, allowing a disulphide link, with a C-terminal hexa-His-tag on MacB and pRSFDuet-TolC-FLAG expressing full-length TolC with a C-terminal FLAG-tag (*Fitzpatrick et al., 2017*). Transformed cells were spread onto LB agar plates containing 50 µg mL⁻¹ carbenicillin and 50 µg mL⁻¹ kanamycin and were incubated at 37°C overnight. A single colony was used to inoculate 50 mL of LB medium containing 50 µg mL⁻¹ carbenicillin and 50 µg mL⁻¹ kanamycin. The preculture was grown at 37°C, with an orbital shaker set at 200 rpm for 4 hr. When the absorbance at 600 nm reached between 0.5 and 0.7, the preculture was used to inoculate 6 L of 2YT medium containing 50 µg mL⁻¹ carbenicillin and 50 µg mL⁻¹ kanamycin in six 2 L baffled flasks. Bacteria were grown at 37°C, 200 rpm until OD$_{600nm}$ reached 0.5 and were subsequently induced with 0.25 mM isopropyl β-d-1-thiogalactopyranoside (IPTG). The culture was further incubated at 20°C, 200 rpm overnight. Cells were collected by centrifugation (5000×*g* for 30 min at 4°C), and pellets were resuspended in a buffer containing 20 mM Tris-HCl pH 8.0 mM and 400 mM NaCl (buffer A).

One tablet of EDTA-free protease inhibitor per 50 mL, 5 µg mL⁻¹ lysozyme and 5 U mL⁻¹ DNase I were added to the cell suspension and incubated 1 hr at 4°C. The mixture was then passed five times through a water-cooled high-pressure homogeniser (Emulsiflex), keeping the pressure below 14,500 psi throughout. The resulting lysate was centrifuged (9000×*g*, 30 min, 4°C) to remove cell debris and then ultracentrifuged (175,000×*g*, 4 hr, 4°C) to pellet membranes.

Membranes were solubilised in 50 mL of buffer 20 mM Tris pH 8.0, 400 mM NaCl with 1.5% (wt/vol) β-dodecyl maltoside (β-DDM) and one tablet of EDTA-free protease inhibitor for 3 hr at 4°C with gentle stirring. The solution was then purified by Ni-NTA chromatography after addition of 10 mM imidazole. The elution peak was further purified with ANTI-FLAG M2 affinity resin (0.5 mL, Sigma-Aldrich), pre-equilibrated with buffer A supplemented with 0.05% wt/vol β-DDM. The elution fraction was added to the resin and incubated at 4°C for 1 hr with gentle agitation. The sample was then washed with buffer A containing 0.05% β-DDM and then 0.03% (wt/vol) decyl maltose neopentyl glycol. Peptidisc (*Carlson et al., 2018*) at 1 mg mL⁻¹ in 400 mM Tris-HCl and additional *E. coli* lipids (100 µL at 20 mg mL⁻¹) were added to the sample on the resin. After 16 hr at 4°C, the resin was washed with buffer A containing no detergent and then eluted with 3xFLAG peptide (Sigma-Aldrich). Flow-through was collected at each step and analysed by SDS-PAGE. The presence of the full assembly was confirmed by mass photometry (Refeyn).

### MacAB-TolC cryo-EM sample preparation and data collection

For single-particle cryo-EM structure solution, the samples of MacAB-TolC complex reconstituted in peptidisc (*Carlson et al., 2018*) were thawed and diluted to 0.5 mg mL⁻¹ in a buffer composed of 20 mM Tris pH 8.0, 100 mM KCl, 2 mM TCEP, and 1 mM NaN₃. A 3 µL aliquot of dilute protein was applied onto a Cu 300-mesh EM grid layered with R1.2/1.3 holey carbon support film (Quantifoil Micro Tools, GmbH), which had been glow-discharged immediately prior to use. The sample deposited on the EM grid was blotted of excess solution and vitrified in liquid ethane using Vitrobot Mark IV (Thermo Fisher Scientific, Inc) robotic cryo-EM plunger.

The grids were imaged using a Titan Krios G3 transmission electron microscope (Thermo Fisher Scientific Inc) operating at an accelerating voltage of 300 kV and liquid nitrogen temperature. The images were recorded in EFTEM mode using AFIS on K3 direct electron detector (Gatan, Inc) and data acquisition software EPU (Thermo Fisher Scientific, Inc). Exposures were for 1 s and were collected as 40-frame movies using total electron fluency (dose) of 49.75 e$^-$ Å$^{-2}$ and defocus values between –2.4 μm and –1.0 μm. The movies were collected in super-resolution mode with nominal pixel size 0.415 Å, but were binned 2× following exposure, yielding effective pixel size of 0.83 Å px$^{-1}$.

## MacAB-TolC cryo-EM data processing

The cryo-EM data were processed using cryoSPARC 4.5 software (Structura Biotechnology, Inc). Patch motion correction was used to correct for beam-induced motion, and Patch CTF was used to calculate the CTF. Circular Gaussian blob picking with a min/max diameter of 120/380 Å was used to extract an initial set of pixels in 878 px boxes Fourier-cropped to 600 px (effective pixel size 1.21 Å px$^{-1}$). After 2D classification, these were used as input to a template-picking model and particles re-extracted at 600 px and Fourier-cropped to 400 px (effective pixel size 1.245 Å px$^{-1}$). This set of particles was pruned again with another round of 2D classification and then subjected to ab initio reconstruction. The best volume and particles from 2D classification was used for heterogeneous refinement along with 5 'junk' volumes. The best output was then subjected to non-uniform refinement in $C_1$ symmetry with optimised per-particle defocus, optimised per-group CTF parameters, and fit spherical aberration options used to further improve the refinement. This yielded a 2.74 Å map with 233,835 particles. Finally, local CTF refinement, global CTF refinement, and another round of non-uniform refinement in $C_1$ was performed, followed by reference-based motion correction. The unbinned output particles (233,775 particles at 600 px box size, 0.83 Å px$^{-1}$) were subjected to a final round of non-uniform refinement in $C_3$ symmetry yielding a final map with a GSFSC reported resolution of 2.48 Å. To facilitate model building, Resolve from Phenix package was also run on this map. Data collection and processing parameters are shown in *Supplementary file 1*.

## MacAB-TolC-YbjP model building and refinement

The part of the experimental map containing an unidentified protein density adjacent to TolC trimer was visualised and traced as a poly-alanine chain using Coot 0.9 (*Emsley et al., 2010*). The resulting structure model was submitted to the DALI server (*Holm, 2022*) to query the *E. coli* species subset of the AlphaFold Database (Hierarchical AF-DB Search). The search showed strong structural homology with the uncharacterised lipoprotein YbjP (AF-P75818-F1). The AlphaFold model was fitted into the experimental map and manually adjusted in Coot 0.9, allowing the identification of the density as YbjP. Following minor structure adjustments, the model was refined using ISOLDE 1.6 plugin (*Croll, 2018*) for ChimeraX-1.6.1 (*Pettersen et al., 2021*). Molecular refinement was completed in Phenix-1.20 (*Liebschner et al., 2019*). Refinement statistics are listed in *Supplementary file 1*.

## Plasmid construction and purification of soluble YbjP

Soluble YbjP bearing a thrombin-cleavable N-terminal His-tag was constructed by amplifying the protein globular domain (residues 28–171) from *E. coli* MG1655 genomic DNA using primers P1 and P2 (*Supplementary file 7*). The PCR product was digested with NdeI-BamHI prior to insertion into pET28a, digested with the same enzymes. *E. coli* BL21 (DE3) cells were then transformed with pET28-YbjP$_{28-171}$ plasmid encoding YbjP$_s$. Bacteria were grown at 37°C in 2YT media supplemented with kanamycin and protein expression induced with addition at mid- to late-log phase of 0.1 mM IPTG. After 16 hr at 18°C, cells were harvested by centrifugation (4000×*g*, 10 min), and pellets resuspended in 50 mM Tris pH 7.5, 200 mM NaCl, and 10% glycerol supplemented with lysozyme and DNase I. Cells were then lysed by passage through a cell disruptor (Constant Systems) at 30,200 psi. Unbroken cells and debris were removed by ultracentrifugation (107,000×*g*, 1 hr, 4°C). The supernatant was supplemented with 20 mM imidazole and loaded onto a 5 mL HisTrap FF column (GE Healthcare), which was washed with 15 column volumes of the same buffer before elution with 250 mM imidazole. Protein fractions were pooled, dialysed twice against buffer containing 50 mM Tris pH 7.5 mM and 200 mM NaCl, and concentrated using a Vivaspin filter with a 10 kDa MWCO before flash freezing and storage at –80°C.

## Preparation of AcrABZ-TolC in peptidisc reconstituted with YbjP

Co-expression of the recombinant AcrABZ-TolC-3xFLAG and cellular membrane preparation was as previously described with some modifications (*Du et al., 2014*). The AcrABZ-TolC-3xFLAG components were expressed into *E. coli* strain C43 (Δ*acrAB* and Δ*tolC*). Cell pellets were resuspended in lysis buffer (150 mM NaCl and 50 mM Tris pH 7.5) and membranes pelleted as previously described (*Du et al., 2014*) and solubilised with a buffer containing 400 mM NaCl, 50 mM Tris pH 7.5 and 1.5% (wt/vol) β-DDM by stirring overnight at 4°C. The soluble extract was incubated with Ni-NTA beads (QIAGEN) for nickel affinity purification (2 mL for 4.5 g membrane extract) of the His-tagged AcrABZ$_{6xHis}$-TolC$_{3xFLAG}$. The beads were loaded onto a gravity flow-column and the column washed with 4 column volumes (CVs) of a buffer composed of 200 mM NaCl, 50 mM Tris pH 7.5, and 0.02% (wt/vol) β-DDM. The AcrABZ$_{6xHis}$-TolC$_{3xFLAG}$ protein complex was eluted with the same buffer containing 0.5 M imidazole. Eluted fractions were analysed on a 4–12% SDS-PAGE gel, and fractions enriched for the pump were then pooled for a second purification step using anti-FLAG M2 affinity beads (Sigma). The fractions were incubated with 0.5 mL anti-FLAG resin for 3 hr at 4°C with gentle stirring, and the beads were then washed with 2 CVs of buffer (200 mM NaCl, 50 mM Tris pH 7.5 and 0.02% β-DDM) and 3 CVs of buffer 200 mM NaCl, 50 mM Tris pH 7.5 to gradually decrease the detergent of the solubilised pump to below the critical micelle concentration. For reconstitution of the AcrABZ-TolC sample, the NSPr peptidisc was used based on the protocol by *Carlson et al., 2018*, with some modifications. Next, 2 mg mL$^{-1}$ of peptidisc (Peptidisc Biotech) in 20 mM Tris-HCl pH 8.0 peptide solution was loaded to the column and incubated for 20 min at room temperature with gentle shaking. The flow-through was collected by gravity, and 2 mg mL$^{-1}$ peptide solution containing 40 µL of *E. coli* lipids (6 mg mL$^{-1}$) was loaded to the column and incubated with the beads. Finally, 0.5 mg mL$^{-1}$ of 3xFLAG peptide in buffer 50 mM Tris pH 7.5, 200 mM NaCl was added to the column and incubated with the beads for 15 min. The fractions were analysed on a 4–12% gradient SDS-PAGE gel and concentrated to 5–6 mg mL$^{-1}$ using a Vivaspin filter with a 100 kDa MWCO. Prior to grid freezing, samples of AcrABZ-TolC in peptidisc were mixed with 10-fold excess of C-terminally His-tagged soluble YbjP.

## AcrABZ-TolC-YbjP cryo-EM sample preparation, data collection, and processing

Sample of AcrABZ-TolC in peptidisc and YbjP was applied on a Cu 300-mesh EM grid layered with R1.2/1.3 holey carbon support film (Quantifoil), which had been glow-discharged immediately prior to use. The sample deposited on the EM grid was blotted of excess solution and vitrified in liquid ethane using Vitrobot Mark IV (Thermo Fisher Scientific, Inc) robotic cryo-EM plunger. Images were collected on a Titan Krios microscope operating at 300 kV. Electron fluency was 56 e$^{-}$ Å$^{-2}$ and defocus values between –2.5 µm and –1.0 µm. Data were processed with cryoSPARC as detailed in *Figure 4— figure supplement 1*. After patch motion correction and CTF correction, particles were selected using Blob picker and 8603 particles were extracted prior to 2D classification and best classes were used for template picking. Particles were then subjected to two rounds of 2D classification. The selected 257,681 particles were used to build three ab initio models. The tripartite pump was further refined by heterogeneous refinement and non-uniform refinement in $C_3$ symmetry, yielding a final map at 3.17 Å resolution.

## AcrABZ-TolC-YbjP preparation with mature peptidoglycan and cryo-EM processing

The AcrABZ-TolC-YbjP prepared in peptidisc were mixed with roughly 1 mg mL$^{-1}$ of peptidoglycan slurry. Samples of mature and early forms of peptidoglycan (PG) were kindly provided by Prof. Waldemar Vollmer (University of Newcastle and now at University of Queensland, Brisbane, Australia), prepared from strains BW25113 ΔGLDT and C5703-1. The mixture was incubated on ice for 1 hr and then used to prepare grids on glow-discharged Ultrafoil R 1.2/1.3 using an FEI Vitrobot (IV) at settings of 95% humidity, 4°C, 3 s blot time, –5 force. Cryo-EM data processing and 3D reconstructions were carried out using cryoSPARC (*Punjani et al., 2017*). Multiple rounds of 3D refinement were conducted, including heterogeneous refinement to remove non-pump particles, along with homogeneous refinement and non-uniform refinement to improve the resolution, yielding a final map of AcrABZ-TolC-PG-YbjP at 3.97 Å. The data collection and processing parameters are summarised in *Supplementary file 2*.

## Isothermal titration calorimetry

Experiments were performed with a ITC200 calorimeter (MicroCal, Malvern Panalytical) at 25°C. After overnight dialysis in 20 mM HEPES pH 7.5, 200 mM NaCl, 0.03% (wt/vol) DDM, N-terminally His-tagged soluble YbjP at 230 µM was injected into 40 µM TolC in the same buffer. The titration programme consisted of 25 injections of 1.5 µL with an initial injection of 0.4 µL every 300 s, with cell stirring rate of 750 rpm. For each condition, a control titration was performed by injecting YbjP into a cell containing buffer only. The control signal was then subtracted from the experimental titration, and the binding affinity, stoichiometry, and thermodynamic parameters were obtained by nonlinear least-squares fitting of the background-corrected experimental data using a single-site binding model from the MicroCal PEAQ-ITC analysis software (Malvern). ITC parameters are listed in *Supplementary file 3*.

## Immobilised metal affinity chromatography binding assay of TolC and YbjP

Detergent-purified FLAG-tagged TolC was mixed to N-terminally His-tagged $YbjP_s$ for 5 min in 50 mM Tris pH 7.5, 200 mM NaCl, and 0.03% (wt/vol) β-DDM. Each protein was at 15 µM final concentration in a final volume of 125 µL. A control was performed with only the TolC protein, or only the YbjP, or by replacing YbjP with another periplasmic protein, soluble LolB, produced as described in *Kaplan et al., 2018*. Proteins were added to 100 µL of Ni-resin (50% slurry, profinity Bio-Rad) in a microbatch spin column and after 5 min the flow-through was recovered. The resin was washed four times with 250 µL of buffer, and bound proteins were eluted with the same volume of buffer containing 250 mM imidazole. Elution fractions were analysed on a gradient (4–12%) SDS-PAGE gel. As a reference, the TolC flow-through fraction was also loaded.

## In vivo photo-crosslinking

### Plasmid constructions

Full-length YbjP was amplified from *E. coli* MG1655 genomic DNA using P3 and P4 primers (*Supplementary file 7*). The PCR product was digested by NdeI-Xho and inserted into pET24a digested by the same enzymes. AcrA was cloned as described for YbjP using the primers P15 and P16. Amber codons were introduced at desired positions by Quikchange site-directed mutagenesis using the primers P5 to P14 (YbjP) and P17 to P20 (AcrA).

### Culture and photo-crosslinking

*E. coli* C43 cells were co-transformed with pEVOL encoding an amber suppressor tyrosyl tRNA and the engineered tyrosyl tRNA synthetase (*Chatterjee et al., 2014*) and with YbjP (pET24-YbjP$_{FL}$-XnTAG) or AcrA (pET24-AcrA-XnTAG) amber mutant for incorporation of pBPA into the TAG amber codon at desired positions. Cultures of 20 mL LB media, supplemented with 20 µg mL$^{-1}$ and 50 µg mL$^{-1}$ of chloramphenicol and kanamycin, respectively, were inoculated with 1/100 of overnight cultures and grown at 30°C. When an optical density of 0.3 was reached, 1 mM of freshly prepared pBPA (Bachem) in NaOH was added in each culture with additional HCl to balance pH. When indicated, a control without addition of pBPA was carried out. After 30 min at 30°C, cells were induced with 0.02% (wt/vol) L-arabinose and 0.5 mM IPTG and grown for 2 hr at 30°C. Cells corresponding to 4 mL of culture were collected by centrifugation (4000×*g*, 4°C, 4 min), washed with 1 mL of PBS, and resuspended in 700 µL PBS. Aliquots of 150 µL of each culture were transferred in a 96-well plate and exposed under UV (365 nm) for 15 min at room temperature using a UVGL-58 handheld UV lamp.

### His-tag enrichment

The next day, samples were lysed by three cycles of freezing in liquid nitrogen and thawing in 37°C water bath. Lysozyme at 0.5 mg mL$^{-1}$, DNase at 50 µg mL$^{-1}$, and 1% (wt/vol) DDM were added. After 30 min at room temperature, samples (300 µL) were transferred into spin columns containing 100 µL of Ni-NTA resin (50% suspension, profinity Bio-Rad) previously equilibrated with PBS (phosphate buffered saline). Proteins were allowed to bind for 5 min at room temperature. The resin was then washed three times with 400 µL of PBS supplemented with 0.03% DDM and bound proteins eluted with 150 µL of PBS containing 0.03% DDM and 250 µM imidazole.

## Western blot

Samples (20 μL) were analysed by SDS-PAGE and immunoblotting with mouse anti-His primary antibodies (Penta-His, QIAGEN) and goat anti-mouse secondary antibodies (IRDye 800 CW, LI-COR) prior to imaging with an Odyssey LI-COR system. For the YbjP and TolC double detection (*Figure 3—figure supplement 1*), proteins were simultaneously detected with mouse anti-His and rabbit anti-TolC primary antibodies and with goat anti-mouse secondary antibodies (IRDye 800 CW, LI-COR) and donkey anti-rabbit secondary antibodies (IRDye 680 RD, LI-COR).

## Strains and culture conditions

*E. coli* BW25113 wild-type and Δ*tolC* were obtained from the Keio collection (*Baba et al., 2006*). BW25113 Δ*tolC* Δ*ybjP* was generated by $\lambda$-Red recombination from the Δ*tolC* strain (*Supplementary file 7*). Due to consistent deletion of the *yecT*- *flhD* region of the Keio collection BW25113 Δ*ybjP* strain following FLP expression, Δ*ybjP* was generated by $\lambda$-Red recombination from the wild-type strain. The kanamycin-resistance cassettes were excised from these strains using pCP20, and the genomes were sequenced for confirmation.

Cells were cultured routinely in Mueller Hinton (MH) broth at 37°C, with shaking at 200 rpm. Where specified, cells were cultured in Luria-Bertani (LB) broth (0.5 g $L^{-1}$ NaCl) or M9 minimal media supplemented with 0.4% glycerol. Cells containing a gentamicin-resistance version of the pKD46 plasmid were cultured in LB broth containing 10 μg $mL^{-1}$ gentamicin, except Δ*tolC*, which was cultured with 5 μg $mL^{-1}$ gentamicin. LB agar plates containing 50 μg $mL^{-1}$ kanamycin or 50 μg $mL^{-1}$ carbenicillin were used for routine transformant selection. Genome-integrated kanamycin-resistance cassettes were selected on LB agar containing 30 μg $mL^{-1}$ kanamycin, except for Δ*ybjP* and Δ*tolC* Δ*ybjP*, which were plated on LB agar containing 20 μg $mL^{-1}$ and 10 μg $mL^{-1}$ kanamycin, respectively.

## $\lambda$-Red recombination

$\lambda$-Red recombination (*Datsenko and Wanner, 2000*) was carried out using a modified version of the protocol described in *Peng et al., 2022*. Briefly, cultures of cells containing a gentamicin-resistance version of the pKD46 plasmid were prepared in LB broth containing gentamicin and incubated overnight at 30°C. The overnight cultures were diluted 1:200 into fresh LB broth containing gentamicin and incubated at 30°C until the $OD_{600nm}$ reached 0.4, at which point L-arabinose was added to a final concentration of 0.3% (wt/vol), and the cultures were incubated at 37°C for 1 hr. The cultures were transferred to ice, then washed by four cycles of centrifugation and resuspension in ice-cold distilled water. After the final centrifugation, the distilled water was discarded, and the cells were resuspended in the residual liquid volume. The $OD_{600nm}$ of the resuspended cells was measured, and appropriate volumes of ice-cold distilled water were added to bring the cells to a final density of $OD_{600nm}$ 50.

Amplicons of kanamycin-resistance cassettes with 50 bp homology to the gene of interest were designed as described in *Baba et al., 2006*, amplified using Q5 polymerase (NEB), and gel purified (NEB). 100 ng of each purified amplicon was added to 70 μL of the pKD46-induced electrocompetent cells and transformed by electroporation. Cells were recovered in SOC media (NEB) at 37°C for 1 hr, then plated on selective agar plates. The kanamycin-resistance cassettes were excised from the strains using the plasmid pCP20, as described in *Baba et al., 2006*.

## Antimicrobial susceptibility testing

Minimum inhibitory concentrations (MICs) were determined by broth microdilution. Briefly, overnight cultures in MH broth were diluted 1:1000 in fresh MH broth and incubated at 37°C until reaching an $OD_{600nm}$ of 0.2–0.5. The cultures were then diluted to an $OD_{600nm}$ of 0.001 in fresh MH broth, and 50 μL were added to wells of flat-bottom 96-well microtiter plates already containing 50 μL of twofold serially diluted compounds in MH broth, to a final volume of 100 μL. The plates were sealed with a gas-permeable seal and incubated at 37°C for 18 hr. Growth was determined by visual inspection and $OD_{600nm}$ measurement using a SpectraMax iD3 microplate reader.

For antimicrobial susceptibility testing under microaerobic conditions, prior to incubation, the 96-well microtiter plates were placed in a 2.5 L anaerobic jar and Millipore Anaerocult C was used following the manufacturer's instructions to generate an environment containing 5–6% $O_2$ and 8–10% $CO_2$.

## Bacterial growth curves

Bacterial growth curves were carried out by diluting overnight cultures to an $OD_{600nm}$ of 0.02 in fresh media containing the relevant concentrations of supplemented solutions. 200 µL of the diluted cultures were added to wells of a flat-bottom 96-well microtiter plate and the plate was sealed with a gas-permeable seal. $OD_{600nm}$ was read every 10 min for 24 hr at 37°C with shaking using a SpectraMax iD3 microplate reader.

## Swimming motility assay

LB motility agar plates were prepared by adding 25 mL of LB broth containing 0.3% agar in 94 mm diameter sterile Petri dishes. Overnight cultures of the relevant strains were prepared in LB broth. A volume of 2 µL of overnight culture was injected into the centre of each motility agar plate, followed by incubation at 37°C for 24 hr. The diameters of the colonies at the widest point were measured using a ruler.

## Proteomics sample preparation and analysis

Cultures were prepared in triplicate under aerobic conditions in MH media for the four strains Δ*ybjP*, Δ*tolC*, Δ*tolC* Δ*ybjP*, and the BW25113 parent, all prepared by the recombineering approach described above. Cultures were grown to exponential phase (0.6–0.8 $OD_{600\,nm}$, 15 mL) and to stationary phase (4.8–5 $OD_{600\,nm}$, 5 mL) and pelleted by centrifugation at 4000×$g$, 5 min and frozen. The pellets were resuspended in 1 mL lysis buffer for set 1 (50 mM Tris-Cl pH 7.5, 100 mM NaCl, 1 mM $MgCl_2$, 1.4 mg $mL^{-1}$ hen egg white lysozyme, 1% (vol/vol) Triton X-100). The suspension was sonicated with a micro-probe tip (Sonics vibracell) on ice for 60 s, with 4 s pulses, 6 s wait time, 50% amplitude. To the soni-cate lysate, 2 units of DNase I were added to digest DNA, followed by incubation on ice for 10 min. Trichloroacetic acid (Sigma) was then added to 5% (wt/vol), then incubated on ice and collected by centrifugation in an Eppendorf centrifuge (13,000 rpm, 5°C, 10 min). The pellet was washed with 5% TCA in water, then spun (set 1) or with 50% ethanol in 20 mM Tris-Cl pH 7.5 (set 2), and the collected pellet provided to the University of Cambridge Centre for Proteomics. Samples were digested with trypsin and analysed by quantitative label-free LC-MS/MS. Signals from total unique spectral counts were normalised for each replicate and averaged.

Raw mass spectrometry data was processed with DIA-NN v2.1.1 (*Demichev et al., 2020*) using the reference *E. coli* proteome from UniProt (acc. no. UP000000625) along with contaminant protein sequences provided by the Cambridge Centre for Proteomics (*Dawson and Smith, 2024*). Precursor and fragment mass tolerances were automatically optimised within the DIA-NN search. Trypsin was set as the enzyme of choice, and a maximum of one missed cleavage was allowed. Carbamidometh-ylation of cysteine was set as a fixed modification, one variable modification was allowed from either N-terminus acetylation or oxidation of methionine. An FDR cutoff of 1% was used at the precursor level.

Quantitative analysis was performed from the abundances at the peptide level using qfeatures (*Gatto and Vanderaa, 2025*) and limma (*Ritchie et al., 2015*), adapting the code from *Hutchings et al., 2025*. Briefly: peptides matching contaminant proteins, peptides with q-values higher than 0.01, ambiguously mapping peptides and peptides with a proportion of missing values greater than 0.5 across all samples were removed from the analysis. Missing values were imputed using minDet imputation for missing values not at random, and knn for missing values at random. Peptide quan-titation information was aggregated at the protein level using qfeatures with the function colMe-dians, quantitation information was $log_2$ transformed and normalised. Protein-wise linear models were built with limma which were subsequently adjusted with the eBayes function. Differentially expressed proteins were considered if they had $log_2$ fold changes (LFC) greater than 1 or smaller than –1, and statistically significant proteins were considered if their Bonferroni-Hochberg adjusted p-value was smaller than 0.05.

## Phylogenetic analysis of DUF3828 domain

3327 proteins containing DUF3828 domain (PF12883) were acquired from InterPro (v103), and the domain architectures represented by *E. coli* YbjP and YqhG were selected. Taxonomy information was retrieved from UniProt on 22 December 2024, and the signal peptide was predicted by SignalP 6.0 (*Teufel et al., 2022*). After removing proteins with unidentified taxonomy, representative sequences

were selected by manual inspection of *E. coli* and *Salmonella* and CD-HIT with a 95% identity threshold (*Fu et al., 2012*), yielding 356 sequences. The multiple sequence alignment was generated by hmmalign using the PF12883 HMM profile and processed with TrimAl on automated mode (*Capella-Gutiérrez et al., 2009*). A phylogenetic tree was constructed from the resulting multiple sequence alignment using IQ-Tree2 on a default parameter (*Nguyen et al., 2015*). The tree was rooted using the minimal ancestor deviation method (*Tria et al., 2017*), yielding the same result. Annotation was performed with TreeViewer (*Bianchini and Sánchez-Baracaldo, 2024*), and webFlaGs (*Saha et al., 2021*) was used to analyse the gene synteny of DUF3828-containing proteins.

### Bioinformatic analysis of outer membrane efflux protein

Outer membrane efflux proteins (IPR003423) and TolC proteins (IPR010130) were acquired from InterPro (v103), and the taxonomic information was retrieved from UniProt on 22 December 2024. The signal peptide was predicted by SignalP6.0. Outer membrane efflux proteins of *Pseudomonadota* were selected and clustered with CLANS (*Frickey and Lupas, 2004*).

For the phylogenetic analysis of TolC, the method described for DUF3828 analysis was used with some modifications. Sequences of *Pseudomonadota* were collected, and the sequence showing the highest sequence similarity to *E. coli* TolC in each organism was selected. Redundant sequences were removed by CD-HIT with a 90% identity threshold. In addition, a phylogenetic tree was constructed using FastTree on -mlacc 2 -slowni parameter (*Price et al., 2010*).

### Materials availability

Materials are available from the corresponding author.

### Acknowledgements

JG was supported by an Amgen scholarship. Grids were prepared and cryo-EM data collected at the BIOCEM facility, Department of Biochemistry, University of Cambridge. We thank Dimitri Y Chirgadze, Steven Hardwick, and Lee Cooper for assistance with data collection and processing at the Cryo-EM Facility. We thank Yulia Yuzenkova for providing samples of microcin J25 and Martin Pos for TolC antibodies and helpful discussions. We thank Somenath Bakshi, Liz Sockett, and Waldemar Voller for helpful discussions, and Waldemar for providing mature and early forms of peptidoglycan. This work is supported by an ERC Advanced Award (742210) and Wellcome Trust Investigator Awards (222451/Z/21/Z and 200873/Z/16/Z). AZ is a recipient of a Transition To Independence (TTI) fellowship from the School of Biological Sciences at the University of Cambridge, supported by funding from the Rosetrees Trust (JS16/TTI2021\1) and the Isaac Newton Trust (21.22(a)iii) and the School of Biological Sciences at the University of Cambridge.

### Additional information

#### Funding

| Funder | Grant reference number | Author |
| --- | --- | --- |
| Wellcome Trust | 10.35802/222451 | Jim Horne<br>Elise Kaplan<br>Ben Jin<br>Emmanouela Petsolari<br>Jan Gradon<br>Yvette Ntsogo<br>Andrzej Harris<br>Dingquan Yu<br>Ben F Luisi |
| European Research Council | 742210 | Jim Horne<br>Elise Kaplan<br>Yvette Ntsogo<br>Ben F Luisi |

| Funder | Grant reference number | Author |
|---|---|---|
| School of the Biological Sciences, University of Cambridge | Transition to Independence | Ashraf Zarkan |
| Rosetrees Trust | JS16/TT12021\1 | Ashraf Zarkan |
| Isaac Newton Trust | 21.22(a)iii | Ashraf Zarkan |

The funders had no role in study design, data collection and interpretation, or the decision to submit the work for publication. For the purpose of Open Access, the authors have applied a CC BY public copyright license to any Author Accepted Manuscript version arising from this submission.

## Author contributions

Jim Horne, Data curation, Formal analysis, Supervision, Investigation, Visualization, Methodology, Writing – original draft; Elise Kaplan, Conceptualization, Data curation, Formal analysis, Supervision, Validation, Investigation, Visualization, Methodology, Writing – original draft, Project administration, Writing – review and editing; Ben Jin, Formal analysis, Investigation, Visualization, Methodology, Writing – original draft, Writing – review and editing; Kieran Abbott, Data curation, Formal analysis, Investigation, Methodology, Writing – original draft, Writing – review and editing; Victor Flores, Data curation, Formal analysis, Visualization, Methodology, Writing – original draft, Writing – review and editing; Emmanouela Petsolari, Data curation, Formal analysis, Investigation, Visualization; Jan Gradon, Data curation, Formal analysis, Investigation, Visualization, Methodology, Writing – original draft, Writing – review and editing; Yvette Ntsogo, Formal analysis, Investigation; Andrzej Harris, Formal analysis, Visualization, Methodology; Dingquan Yu, Formal analysis, Investigation, Methodology; Ashraf Zarkan, Data curation, Supervision, Funding acquisition, Investigation, Methodology, Writing – original draft, Project administration, Writing – review and editing; Ben F Luisi, Conceptualization, Formal analysis, Supervision, Funding acquisition, Investigation, Visualization, Methodology, Writing – original draft, Project administration, Writing – review and editing

## Author ORCIDs

Jim Horne ⓘ https://orcid.org/0000-0001-5260-2634
Elise Kaplan ⓘ https://orcid.org/0000-0003-4985-1482
Ben Jin ⓘ https://orcid.org/0009-0006-3491-9404
Kieran Abbott ⓘ https://orcid.org/0009-0006-3369-5920
Victor Flores ⓘ https://orcid.org/0000-0003-4186-9151
Andrzej Harris ⓘ https://orcid.org/0000-0003-4885-7401
Ben F Luisi ⓘ https://orcid.org/0000-0003-1144-9877

Reviewer #2 (Public review): https://doi.org/10.7554/eLife.110666.3.sa1
Author response https://doi.org/10.7554/eLife.110666.3.sa2

# Additional files

## Supplementary files

Supplementary file 1. Cryo-EM data and refinement statistics for peptidisc-reconstituted MacAB-TolC-YbjP and AcrABZ-TolC-YbjP models.

Supplementary file 2. Thermodynamic parameters for the TolC-YbjP soluble interaction.

Supplementary file 3. Cryo-EM data collection statistics for AcrABZ-TolC-YbjP in peptidoglycan.

Supplementary file 4. Criteria for classification of DUF3828-containing proteins.

Supplementary file 5. Minimum inhibitory concentration (MIC) values for wild-type and indicated knockout strains of *E. coli* BW25113.

Supplementary file 6. Colony diameters of wild-type and knockout strains of *E. coli* BW25113. Values are presented as mean ± standard deviation from at least 5 replicates.

Supplementary file 7. List of primers for PCR amplification.

MDAR checklist

## Data availability

The MacAB-TolC-YbjP and AcrABZ-TolC-YbjP atomic models and electron microscopy maps have been deposited in the Protein Data Bank (PDB) and Electron Microscopy Data Bank (EMDB) under the accession codes 9QGY, EMDB-53150, and 9TG4, EMD-55890 respectively. Proteomics data have been deposited with the PRIDE database with identification code PXD070953.

The following datasets were generated:

| Author(s) | Year | Dataset title | Dataset URL | Database and Identifier |
| --- | --- | --- | --- | --- |
| Kaplan E, Horne J, Luisi BF | 2026 | Structure of the YbjP lipoprotein bound to the MacAB-TolC tripartite efflux pump | https://doi.org/10.2210/pdb9QGY/pdb | Worldwide Protein Data Bank, 10.2210/pdb9QGY/pdb |
| Kaplan E, Harris A, Horne J, Petsolari E, Luisi B | 2026 | Structure of the YbjP lipoprotein bound to the AcrABZ-TolC efflux pump | https://doi.org/10.2210/pdb9TG4/pdb | Worldwide Protein Data Bank, 10.2210/pdb9TG4/pdb |
| Luisi B | 2026 | A lipoprotein partner for the *Escherichia coli* outer membrane protein TolC | https://www.ebi.ac.uk/pride/archive/projects/PXD070953 | PRIDE, PXD070953 |

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
