## [Editor Report · eLife Assessment]

In this **fundamental** work Horne et al present **compelling** evidence that YbjP is a novel binding partner of the TolC channel protein. The YbjP is characterized using cryo-EM, and its role probed using pull-down experiments, in vivo crosslinking, functional assays along with phylogenetic analysis which are all properly performed and presented and support the main conclusions. While the study does not identify a clear role for this protein, the revised manuscript offers improved clarity and contributes invaluable insight into membrane transport and antimicrobial resistance.

---

## [Referee Report · Reviewer #2 (Public review)]

This article focuses on the study of two *E. coli* tripartite efflux pumps, both using TolC as a partner in the outer membrane, namely MacAB-TolC and AcrABZ-TolC.

By preparing MacAB-TolC in Peptidiscs rather than in detergent for cryo-EM structure determination, they visualized an extra protein localized around TolC. The resolution was sufficient to build part of the structure, and using the AlphaFold2 database and DALI topology recognition program, they identified it as the lipoprotein YbjP. This protein has an anchorage in the outer membrane, and it was suggested that it could act as a support for TolC, which is the only OMF that does not have an N-terminal extension anchored in the outer membrane, which is very puzzling for the community working in this field of research.

Authors used a large number of different approaches to evaluate the importance of YbjP (structure, genomic evolution, microbiology, photocrosslink in vivo, proteomic profile), but did not succeed in finding it a clear role so far, even if it could be important depending on environmental stress. Nevertheless, their results, obtained with extreme rigour, are of main interest for the comprehension of the complexity of such systems and deserve publication.

Comments on revisions:

Thank you for clarifying the points that puzzled me concerning the crosslink experiments. This version does not need further modifications.

---

## [Author Response]

The following is the authors’ response to the original reviews.

**Reviewer #1 (Public review):**
The presentation and especially main-text illustrative material seem to focus disproportionately on MacAB-TolC-YbjP complex, and the AcrABZ-TolC-YbjP is relegated to supplementary data which is somewhat confusing. There is no high-resolution side view of the AcrABZ-TolC-YbjP side-by-side to MacAB-TolC-YbjP which may be helpful to spot parallels and differences in the organisation of the two systems.

This was previously presented in Supplementary Figure S2. However, because the models were shown at a small scale, we have now included the comparison in a main manuscript (Figure 4). This figure presents AcrABZ-TolC-YbjP and MacAB-TolC-YbjP side-by-side, a structural alignment of TolC-YbjP in the two pumps, and close-up views of the interaction interface.

Supplementary Figure 2 may also be better presented in the main text, as it shows specific displacements of residues upon binding of the YbjP relative to the apo-complexes, although this can be left at the authors' discretion.

We added more text to describe the displacements of residues upon YbjP binding: ‘Nonetheless, the side chains of a few residues in TolC, which mainly correspond to positively charged amino acids (R18, R24, K214, R227, R234), reorient to interact with the YbjP lipoprotein partner (Figure 2B).’

**Reviewer #1 (Recommendations for the authors):**
The work is of high quality and requires minimal modifications, which are mentioned as suggestions above and are mostly connected to the illustrative material.One additional suggestion, which is connected to the earlier BioRxiv preprint, the data seen in Fig 6 of the preprint seems to have been edited out from the current version, and perhaps can be included in a revised version, as it seems to support the "rapid adaptation under stress" role for YbjP, which currently is only speculatively mentioned in p.11, line 365 of the manuscript.

We acknowledge that the BioRxiv preprint Figure 6 can support the rapid adaptation under stress role for YbjP. However, upon sequencing the ΔybjP strain from the Keio collection used in the preprint, we identified a large deletion in the yecT-flhD region. We therefore generated a new ΔybjP strain without the yecT-flhD deletion and repeated the experiment. However, the results with the corrected strain did not support the previous conclusion, and these data were consequently removed in the current manuscript.

**Reviewer #2 (Public review):**
In Figure 3C, the experiment performed with AcrA is clear and the extra band appears at the proper size. On the right panel, it is clear that the crosslink doesn't work when pBPA is placed on residues too far from TolC. Only when introduced on N113 or T110 does a band appear.This is in accordance with an interaction in vivo. Nevertheless, 17 + 54 = 71kDa, which is more than the two bands appearing on the gel. This difference in size migration can occur, but it is not clear when looking at Figure S3. In Figure S3a, the purified proteins are highlighted at approximately the expected size (≈20kDa instead of 17 for YbjP and between 56 and 60kDa in two bands for TolC instead of 54kDa). On the right panel, it seems that the bands are present exactly at the same position, instead of an upper band as expected for the crosslinked YbjP-TolC (at 71kDa). It would be clearer if having the control of the same sample without illumination, revealed by anti-TolC, to see the difference.

We thank the reviewer for pointing out this discrepancy. We identified an error in the molecular weight ladder, as one band was missing. This has now been corrected: YbjP migrates just below 17 kDa, consistent with Figure 3C. In addition, we previously reported a size of 54 kDa for TolC, whereas matured TolC, after signal peptide cleavage, is actually 52 kDa.

We believe that the differences in the apparent molecular weight observed in Figures 3A, 3C and S3 (now S2) mainly result from tagging and post-translation modifications.

In Figure 3A, we used the soluble construct His-YbjP_28-1711_ (theoretical M_w_ ~18 kDa), as also done for the controls in Figures 3C and S3 (now S2). However, for the crosslinking samples, we used full-length His-tagged YbjP, which carries a post-translational lipid modification (theoretical M_w_ ~19 kDa, considering the protein lipidation). The presence of the lipid chains alters the migration as this species migrates at ~15 kDa (Fig 3A). Increased hydrophobicity, due here to YbjP lipidation, could accelerate the migration (Emmanuel et al. 2025 FEBS Open Bio).

In Figure 3A, we used the TolC-FLAG whose apparent M_w_ is ~52 kDa, as previously reported (Fig S3, Fitzpatrick et al. 2017). In Figure S3 (now S2), we used His-tagged TolC (theoretical M_w_ 55 kDa) for the control, which migrates above 56 kDa. In the crosslinking samples, however, we detect tag-free, endogenous TolC, with a theoretical M_w_ of ~51 kDa.

In conclusion, the crosslinked complex composed of lipidated FL YbjP (~15 kDa) and endogenous TolC (~51 kDa) would be expected to migrate at ~66 kDa, which is consistent with what is observed in Figures 3C and S3 (now S2).

A second point that could be discussed further is the comparison of the structure of the pump in the presence of the peptidoglycan with the images previously obtained by tomography. It is not totally clear to me if YbjP could have been positioned in these maps.

There is density corresponding to YbjP in the map obtained in the presence of peptidoglycan. To improve clarity, we have specified the location of the peptidoglycan relative to the pumps in the revised Figure 4, and Supplementary Figure S4, together with the position of YbjP. In both figures, the lipoprotein appears distant from the peptidoglycan density.

**Reviewer #2 (Recommendations for the authors):**
In addition, please add explanations in the legend of Figure 3C concerning the structures.

We added the following description of the structures: ‘As shown underneath, AcrA residues Q136 and Y137, proximal to TolC in the structure of the AcrABZ-TolC pump (PDB 5NG5), were replaced by pBPA. For YbjP, the two residues N113 and T110 proximal to TolC in the MacAB-TolC-YbjP complex (PDB 9QGY) and the three residues N43, N90 and H104 distal to TolC were mutated.’

It would be clearer if having the control of the same sample without illumination, revealed by anti-TolC, to see the difference.

As the amount of crosslinked material is low, samples were enriched via His-tag purification of YbjP prior to Western blotting. In the absence of illumination (see sample N113, UV-), no crosslink would be formed, and therefore TolC would not be co-purified.

In addition, some typo errors have been noted.Table S1 minus is missing for the defocus range for AcrABZ-TolC-YbjP.

Thank you for noting the typo. We have added the minus sign.

Table S3, please specify what is N in the legend.

N is the stoichiometry parameter, which is now specified in the table legend.

Line 237, I suppose it has to refer to Figure S6, not S5.

Thank you for noting the error. We have verified the text matches the figures here and in the entire manuscript.

Several errors are present in the legend of Figure 6.No letters are indicated for the different panels; line 841 must be C, F and I; the indicated colors for the differentially expressed proteins do not correspond to the volcano plots.

Thank you for suggesting the improvements for the labels. We have modified the plot accordingly.

Reference Glavier 2020 has been cited as Glacier on line 72.

We have modified the writing accordingly and checked the reference.